# Design and Fabrication of 3.5 GHz Band-Pass Film Bulk Acoustic Resonator Filter

**DOI:** 10.3390/mi15050563

**Published:** 2024-04-25

**Authors:** Yu Zhou, Yupeng Zheng, Qinwen Xu, Yuanhang Qu, Yuqi Ren, Xiaoming Huang, Chao Gao, Yan Liu, Shishang Guo, Yao Cai, Chengliang Sun

**Affiliations:** 1The Institute of Technological Sciences, Hubei Key Laboratory of Electronic Manufacturing and Packaging Integration, Wuhan University, Wuhan 430072, China; yu_zhou@whu.edu.cn (Y.Z.); 2019300003014@whu.edu.cn (Y.Z.); qinwen_xu@whu.edu.cn (Q.X.); quyuanhang@whu.edu.cn (Y.Q.); renyuqi@whu.edu.cn (Y.R.); gaochao96@whu.edu.cn (C.G.); liuyan92@whu.edu.cn (Y.L.); 2School of Physics & Technology, Wuhan University, Wuhan 430072, China; hxmmmm@whu.edu.cn (X.H.); gssyhx@whu.edu.cn (S.G.); 3Wuhan Institute of Quantum Technology, Wuhan 430072, China

**Keywords:** film bulk acoustic resonator, scandium aluminum nitride, effective electromechanical coupling coefficient, radio frequency filter

## Abstract

With the development of wireless communication, increasing signal processing presents higher requirements for radio frequency (RF) systems. Piezoelectric acoustic filters, as important elements of an RF front-end, have been widely used in 5G-generation systems. In this work, we propose a Sc_0.2_Al_0.8_N-based film bulk acoustic wave resonator (FBAR) for use in the design of radio frequency filters for the 5G mid-band spectrum with a passband from 3.4 to 3.6 GHz. With the excellent piezoelectric properties of Sc_0.2_Al_0.8_N, FBAR shows a large $K_{eff}^{2}$ of 13.1%, which can meet the requirement of passband width. Based on the resonant characteristics of Sc_0.2_Al_0.8_N FBAR devices, we demonstrate and fabricate different ladder-type FBAR filters with second, third and fourth orders. The test results show that the out-of-band rejection improves and the insertion loss decreases slightly as the filter order increases, although the frequency of the passband is lower than the predicted ones due to fabrication deviation. The passband from 3.27 to 3.47 GHz is achieved with a 200 MHz bandwidth and insertion loss lower than 2 dB. This work provides a potential approach using ScAlN-based FBAR technology to meet the band-pass filter requirements of 5G mid-band frequencies.

## 1. Introduction

Wireless communication has overcome the limitations of time and distance on our communication and allows us to transfer information quickly [1,2,3,4]. In particular, fifth-generation (5G) systems have introduced new services to support higher data transmission rates of wireless communication [5,6,7]. The proliferation of 5G communications has led to a gradual increase in data transmission bands, with frequencies covered ranging from 2.4 GHz to 5 GHz [8,9]. Higher frequencies and frequency bands have been pursued to improve radio frequency (RF) systems, especially the filtering action of RF filters. As important elements in 5G data transmission, filters based on surface acoustic wave (SAW) resonators are difficult to use at a frequency higher than 3 GHz because of performance degradation at high frequencies [10,11,12]. The IHP SAW can be used in higher frequency bands, but it requires more complex processing of the multi-layered structure, such as film transfer and bonding process [13,14,15]. On the contrary, a bulk acoustic wave (BAW) filter seems a better choice for 5G communication. Most commercially available BAW filters are constructed with film bulk acoustic resonators (FBARs), in which an air cavity is created between the bottom electrode and the carrier wafer. FBARs can obtain a better effective electromechanical coupling coefficient ($K_{eff}^{2}$) and provide a higher quality factor (*Q*) [16,17]. FBARs are preferred for higher-frequency applications due to characteristics such as good electivity and high power-handling.

For FBARs and FBAR-based filters, which are capable of processing the high frequencies of the 5G system, higher-frequency operations need a reduction in the piezoelectric layer thickness [18,19,20], while commercially available AlN-based FBARs are capable of providing high longitudinal sound velocity *v* (11,354 m/s) and low acoustic and dielectric losses. However, studies have attempted to scale the frequency range of resonators. New material, such as scandium (Sc) doping aluminum nitride (Sc_x_Al_1−x_N), has been used in attempts to increase piezoelectric coefficients. Sc_x_Al_1−x_N offers a manner in which to create reconfigurable filters by utilizing its tuning and polarization-switching properties [21]. As the Sc element ratio in the Sc_x_Al_1−x_N increases, there is a significant increase in the piezoelectric coefficient e_33_ and piezoelectric moduli d_33_ [22]. Sc_x_Al_1−x_N performs well in terms of thermal stability [23]. The results of these studies show that at temperatures up to 1000 °C, the Sc_x_Al_1−x_N wurtzite structure is stable, and little element inter-diffusion happens at the Sc_x_Al_1−x_N /Mo interface [24]. According to reports, increasing the scandium doping in aluminum nitride to 40% can boost the piezoelectric coefficient d_33_ by about five times [25]. Giribaldi et al. demonstrated the high applicability of utilizing Sc_0.3_Al_0.7_N for microacoustic technologies in the sub-6G band [26]. Moreira et al. enhanced the $K_{eff}^{2}$ of FBAR to 12.07% by doping 15% scandium into aluminum nitride [27]. Ding R et al. produced an FBAR-based filter with a center frequency of 3.38 GHz with 160 MHz 3 dB bandwidth. The insertion loss of the filter has a minimum of 1.5 dB [28]. Yang Q designed a high-selectivity FBAR filter for the 3.4–3.6 GHz range with interpolation loss of −2.05 dB [29].

In this work, we report the use of a Sc_0.2_Al_0.8_N-based FBAR to design band-pass filters for the 5G mid-band frequencies of 3.4 to 3.6 GHz. Using high-quality *c*-axis orientation Sc_0.2_Al_0.8_N film, we verified that the 20 at.% Sc doped concentration can achieve a high $K_{eff}^{2}$ of 13.1%, which constitutes an improvement over Moreira’s devices. With different order circuit design for ladder-type filters, the results show that the out-of-band rejection and the insertion loss can be adjusted to different specific requirements. Compared with the filters of Ding R and Yang Q et al., in which the in-band interpolation loss was −1.5 dB and −2.05 dB, the FBAR-based filter in this study has a lower insertion loss, at 1.28 dB. With a fabricated series and parallel FBARs, filters with passband from 3.27 to 3.47 GHz are achieved. The proposed Sc_0.2_Al_0.8_N-based FBAR filters show a potential for 5G mid-band applications with further optimized fabrication controls and updated designs.

## 2. Design and Fabrication

We chose piezoelectric film bulk acoustic devices to construct a 5G mid-band (3.4–3.6 GHz) filter with a center frequency of 3.5 GHz. The designed piezoelectric film bulk acoustic resonator is illustrated in Figure 1. The FBAR consists of a sandwich structure with a piezoelectric layer between the top electrode and bottom electrode (TE and BE, respectively). The electrical field between the two electrodes excites the bulk acoustic wave. As shown in Figure 1b, an air cavity is created between the bottom electrode and the substrate to trap the acoustic wave between the electrodes. Figure 1c shows the working principle of the filter based on FBARs. Each resonator in this filter has two resonant frequencies. One is the series resonant frequency *f_s_*, at which the impedance can be very low (Z_min_), and the other one is a parallel resonant or anti-resonant frequency *f_p_*, at which the impedance can be very high (Z_max_). The parallel resonator in the filter is tuned to be worked at a slightly lower frequency, compared to the series resonator, by adding a mass-loading layer on the top electrode. When *f_p_*_2_, representing the parallel resonant frequency of the parallel resonators, is equal to or slightly lower than *f_s_*_1_, representing the series resonant frequency of the series resonators, a passband is formed between the frequencies near *f_s_*_2_ and *f_p_*_1_. As shown in Figure 1c, at the frequency point *f*_1_, the parallel FBAR can be regarded as a short-circuit state, and the signal cannot be passed to the output port. Hence, the *f*_1_ is the left transmission zero point of the filter. For the frequency point *f*_2_, the impedance of the series FBAR is small enough, while the impedance of the parallel FBAR is very large. The circuit is manifested as a channel state, and the signals are basically transmitted to the output port. At the frequency point of *f*_3_, the series FBAR can be regarded as being in a disconnection state, and the signal cannot pass the output port. Therefore, the *f*_3_ is the correct transmission zero point of the filter. For the design of the 5G mid-band (3.4–3.6 GHz) filter with a center frequency of 3.5 GHz, we used the Mason model [30,31,32] to simulate the transmit characteristics of filters. The effective electromechanical coupling coefficient ($K_{eff}^{2}$) of FBAR calculated by Equation (1) should reach a value of about 12% [33,34], which is suitable for the band-pass width of a 5G mid-band (3.4–3.6 GHz) filter. According to the requirements of FBARs, Sc_0.2_Al_0.8_N piezoelectric thin film should be a functional piezoelectric material, and the designed thickness of each layer is shown in Table 1.
(1)$$k_{eff}^{2}=\frac{\pi^{2}}{4}\frac{f_{s}}{f_{p}}\frac{f_{p}-f_{s}}{f_{p}}$$

The FBAR devices were manufactured in an eight-inch wafer, as shown in Figure 2. First, high-resistivity silicon was etched to form the separation walls; these are used to accurately define the cavity and prevent over-etching from damaging the devices. Next, the cavity was filled with SiO_2_, and the excess sacrificial layer was removed using chemical mechanical polishing. Then, an AlN seed layer was deposited as a buffer layer, and the Mo bottom electrode was dual-deposited and patterned. Subsequently, 500 nm thick Sc_0.2_Al_0.8_N film was reactively sputtered. Then, a 100 nm thick top Mo electrode and a 37 nm thick Mo mass-loading layer were deposited and patterned above the structure. Subsequently, 1 μm thick Al was deposited by magnetron sputtering and patterned to define the probing pads. Finally, the cavity filled with SiO_2_ was opened by using inductively coupled plasma (ICP) etching, and then the hydrofluoric acid vapor release cavity was introduced.

## 3. Results and Discussions

Sc_0.2_Al_0.8_N piezoelectric thin films were prepared by magnetron sputtering (SPTS, Sigma fxP system, Newport, UK) with a 20 at.% doped ScAl alloy target. A sputter power of 6 kW and bias power of 160 W were used for the film deposition, under a substrate temperature of 200 °C, with flow rates of N_2_ and Ar of 60 sccm and 20 sccm, respectively. The surface morphology of the as-deposited film was observed by using scanning electron microscope (SEM) as shown in Figure 3a, and it was found that a small amount of Sc precipitated on the surface of the as-deposited piezoelectric films. The concentration of doped Sc in the marked area of Figure 3a was tested as 21.8 at.% using energy-dispersive X-ray spectroscopy. The morphology of the Sc_0.2_Al_0.8_N piezoelectric thin film, measured by atomic force microscopy, is shown in Figure 3b, along with the corresponding Rq surface roughness value of 9.8 nm. The results of X-ray diffraction (XRD) testing of the Sc_0.2_Al_0.8_N piezoelectric thin films are shown in Figure 3c, and the corresponding diffraction angle of the Sc_0.2_Al_0.8_N (002) peak is 2*θ* = 36°, which corresponds to the (002) orientation. The inset shows the rocking curve of the Sc_0.2_Al_0.8_N piezoelectric film, and the full width at half maximum (FWHM) of the rocking curve is 1.73°, indicating that the piezoelectric film has a good *c*-axis orientation.

Figure 4a shows the optical view of the fabricated FBAR. The structure in the middle part of the figure is the resonant region, which is the main operating region of the FBAR. On each side of the resonant region are Mo anchors linked to each other with test ports. Release holes are retained around the resonance area. The space below the resonance region is released through the release holes to form a cavity. A cross-sectional view of the resonant region is shown in Figure 4b. The growth of the different material films can be clearly identified from the figure. The piezoelectric stack consists of Mo/Sc_0.2_Al_0.8_N/Mo with thicknesses of 122 nm/708 nm/186 nm, respectively, and a 27 nm thick AlN seed layer is under the bottom Mo layer. From the results of SEM inspection, it can be found that the thickness of electrodes and piezoelectric layers is different from the design. The variations in thickness may be an error generated by the machining process.

The impedance responses of series and parallel resonators, measured by a Keysight network analyzer (N5222B) connected to a Cascade Microtech GSG probe station, are shown in Figure 5. Table 2 summarizes the relevant measured and extracted parameters of the series and parallel resonators used in the filter. The quality factor (*Q*) of FBARs can be calculated by Equation (2) [33,35], where the *τ*(*f*) is the group delay of *S*_11_. Based on the Sc_0.2_Al_0.8_N film, the prepared resonator reached a $K_{eff}^{2}$ of 13.1%. However, the resonant frequency of both series and parallel resonators are lower than the designed ones shown in Figure 1b, which can be contributed to the thickness deviations of deposited electrodes and piezoelectric layers.
(2)$$Q\left(f\right)=\frac{2\pi f\tau \left(f\right)\left|S_{11}\right|}{1-{\left|S_{11}\right|}^{2}}$$

Figure 6a is a schematic of the ladder-type circuit design of the FBAR filter consisting of series resonators and parallel resonators. To achieve its passband transmit characteristics, an additional Mo mass-loading layer is added to the parallel resonators to make resonant frequencies lower than those of the series resonator. Figure 6b–d is the optical view of fabricated FBAR filters with different orders. Figure 6b is a second-order ladder-type FBAR filter, which includes two series resonators and two parallel resonators. Figure 6c,d show third- and fourth-order ladder-type FBAR filters, respectively.

The measured transmission responses (S_21_) of the FBAR filters with different orders are shown in Figure 7. Figure 7a illustrates how the out-of-band rejection gradually strengthens as the order of the filter increases. However, the insertion loss becomes worse, and the in-band ripple becomes more severe, as shown in Figure 7b. When the order of the filter is two, the out-of-band rejection is around −15 dB, and the minimum insertion loss is around −1 dB. For the fourth-order filter, the low-frequency out-of-band rejection is below −30 dB. Though the high-frequency out-of-band rejection reduces as the frequency increases, the high-frequency out-of-band rejection is still below −20 dB. The minimum insertion loss is around −1.5 dB, and the ripples are more pronounced in the fourth-order filter. What is more, consistent with the resonant performances of fabricated FBARs, the frequency ranges of the passbands of the filters are from 3.27 to 3.47 GHz, with insertion losses less than 2 dB in a 200 MHz passband range. This passband shift can be also attributed to the thicker electrodes and piezoelectric layers compared with the designed parameters [36,37]. To reduce the in-band ripple, two capacitors and two inductors were added to the circuit of the filters, as depicted in Figure 7c. The capacitances of the capacitors are 0.06 pF and 0.03 pF, and the inductances of the inductors are 1 nH and 0.4 nH, respectively. The out-of-band rejection of filters changes slightly, but the in-band ripple is reduced. The in-band interpolation loss of the fourth-order FBAR-based filter was reduced from −1.28 dB to −1.39 dB. This was mainly due to the fact that the addition of more electrical elements increases the insertion loss in the filter, resulting in a slightly lower in-band ripple. Figure 7d shows that the in-band ripple is reduced in all filters. As the filter order increases, out-of-band rejection improves but, inevitably, leads to increased insertion loss [38].

## 4. Conclusions

In this study, we designed ScAlN-based FBAR filters with a passband of 3.4 to 3.6 GHz for 5G mid-band frequencies and fabricated ladder-type filters with different orders. A 20 at.% Sc doping concentration was chosen to meet the requirement of $K_{eff}^{2}$ of 12% for FBARs, and the $K_{eff}^{2}$ of the fabricated Sc_0.2_Al_0.8_N-based FBARs can reach a value of 13.1%. Ladder-type filters of various orders were fabricated, and the S_21_ results show that the out-of-band rejection gradually strengthened as the order of the filter increased. However, the insertion loss worsens, and the in-band ripple becomes more severe. Finally, a passband from 3.27 to 3.47 GHz with a 200 MHz bandwidth and insertion loss lower than 2 dB was obtained. The achieved frequencies of the passband were lower than the designed ones due to thickness deviations during the fabrication process. With increased precision in the thickness control for the film deposition and external circuit element matching, the proposed FBAR filters can be a potential choice for application in the 5G mid-band spectrum.

## Figures and Tables

**Figure 1 micromachines-15-00563-f001:**
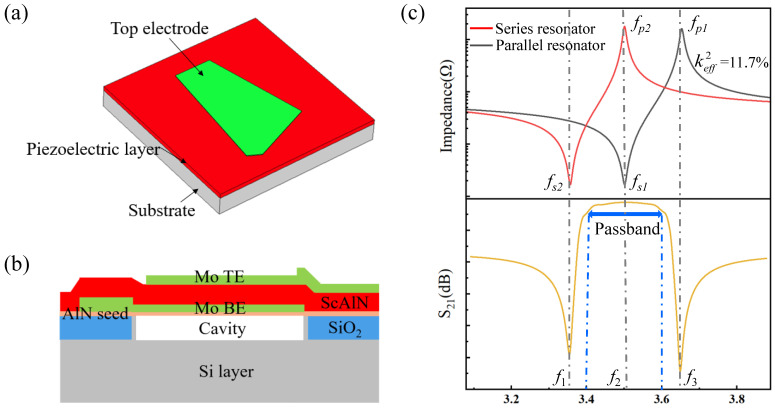
Structures of FBAR and characteristics of filters. (**a**) Schematic drawing of a typical FBAR. (**b**) The cross-sectional view of FBAR. (**c**) Working principle of filter based on FBARs in the circuits model.

**Figure 2 micromachines-15-00563-f002:**
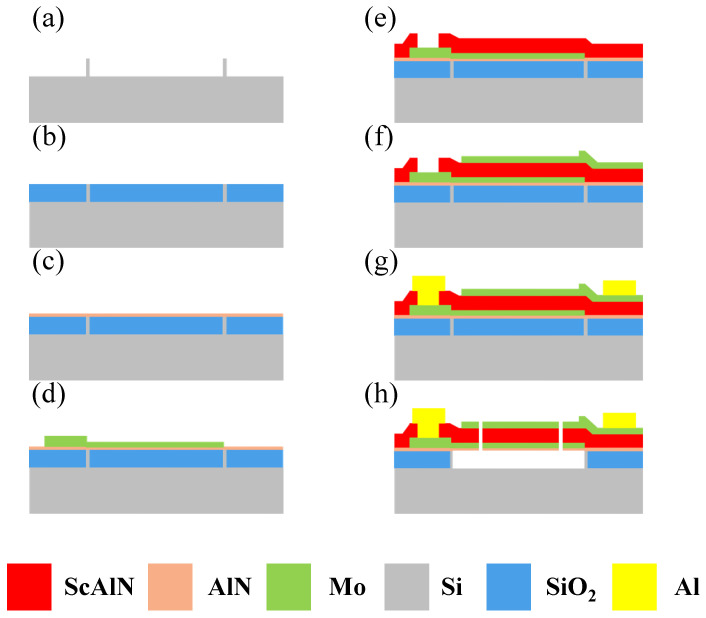
Fabrication of the Sc_0.2_Al_0.8_N-based FBARs. (**a**) The film structure of FBAR. (**b**) Forming separation walls. (**c**) SiO_2_ deposition and chemical mechanical polishing. (**d**) Seed layer deposition. (**e**) Bottom Mo and Sc_0.2_Al_0.8_N layers were dual-deposited and then patterned. (**f**) Top Mo was dual-deposited and then patterned. (**g**) Al was deposited and then patterned. (**h**) Opening the release window and release.

**Figure 3 micromachines-15-00563-f003:**
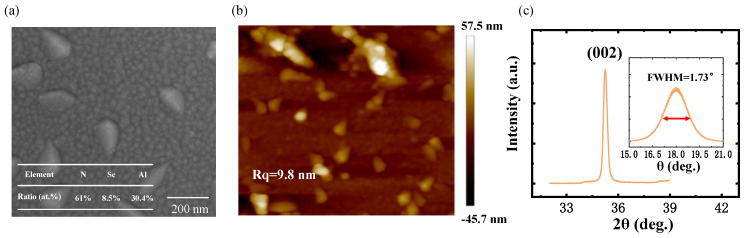
Characterization of the prepared Sc_0.2_Al_0.8_N piezoelectric thin film material. (**a**) The surface morphology observed by SEM. (**b**) The surface morphology observed by AFM. (**c**) XRD results of Sc_0.2_Al_0.8_N.

**Figure 4 micromachines-15-00563-f004:**
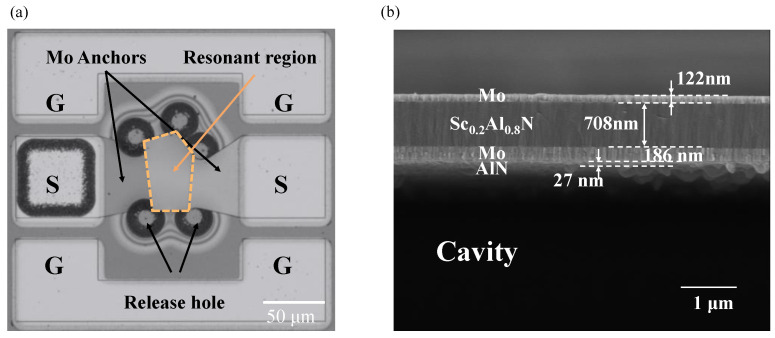
Characteristics of fabricated FBARs. (**a**) SEM image of top view of the fabricated FBAR resonator. (**b**) SEM image of the cross-sectional view of the fabricated FBAR resonator.

**Figure 5 micromachines-15-00563-f005:**
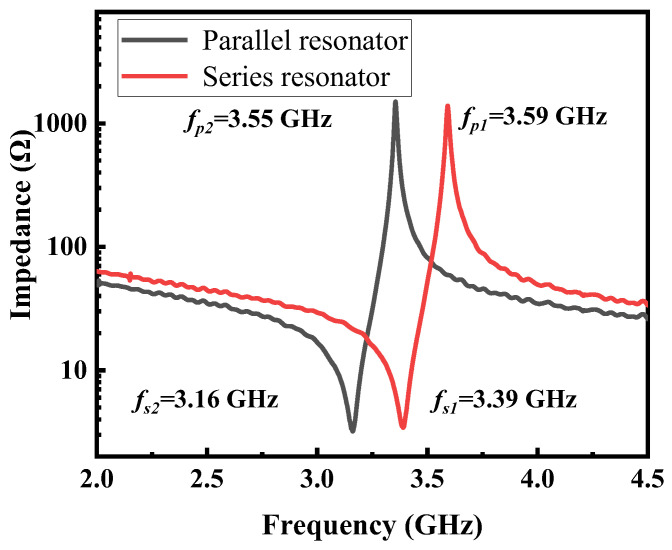
Measured impedance response of series and parallel resonators.

**Figure 6 micromachines-15-00563-f006:**
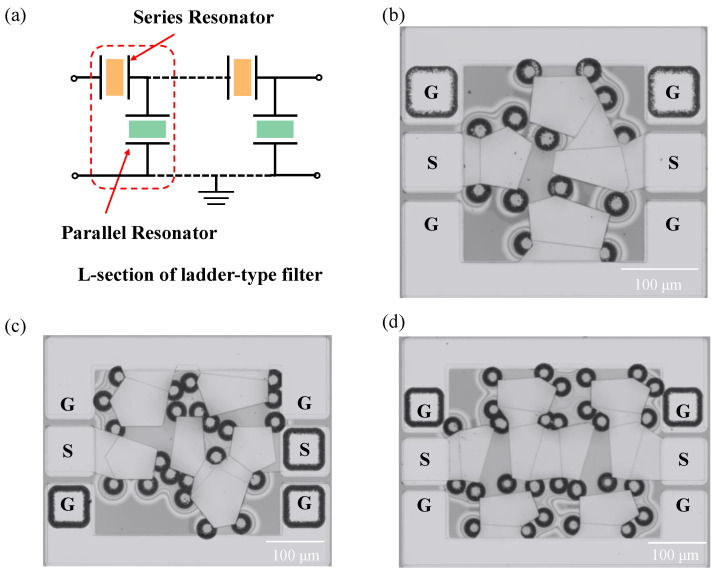
FBAR filters with different topological structures. (**a**) Schematic circuit designs of the FBAR filter. (**b**–**d**) Second-, third- and fourth-order ladder-type FBAR filters, respectively.

**Figure 7 micromachines-15-00563-f007:**
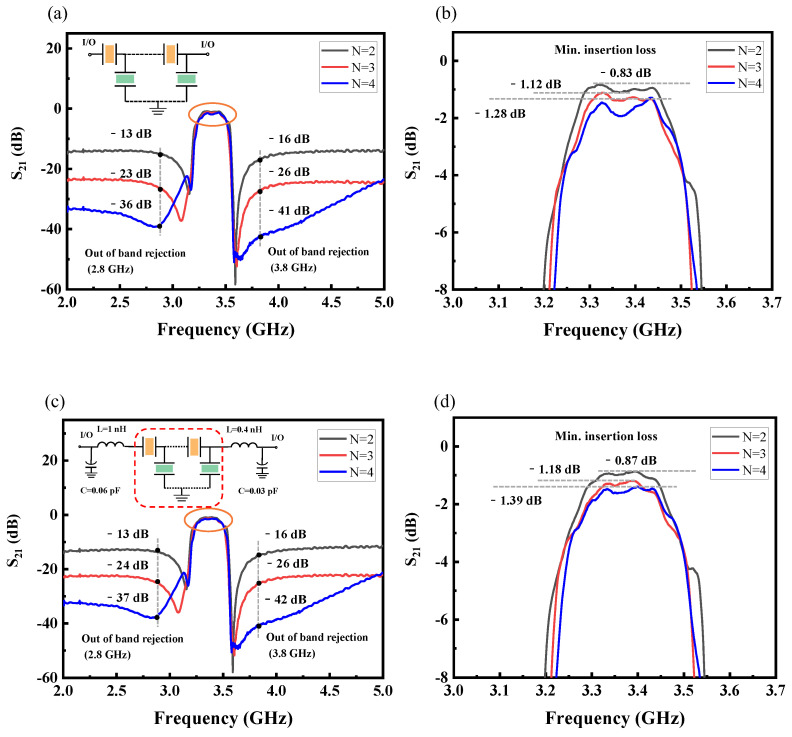
Measured transmission responses (S_21_) of the fabricated FBAR filters. (**a**) The S_21_ responses of the filters with different orders. (**b**) The enlarged view of the passband in (**a**). (**c**) The S_21_ responses of the filters in (**a**) with the external circuit compensation. (**d**) The enlarged view of the passband in (**c**).

**Table 1 micromachines-15-00563-t001:** The parameters of FBAR.

Layer	Thickness (nm)
Mo TE	100
Mo massloading	18
Sc_0.2_Al_0.8_N	700
Mo BE	150
AlN seed	25
Cavity	2000

**Table 2 micromachines-15-00563-t002:** The fabricated resonators used for the FBAR filter.

Resonator	C_0_ (pF)	*f_s_* (GHz)	*f_p_* (GHz)	*Q_s_*	*Q_p_*	$K_{\mathrm{eff}}^{2}$ (%)
Series	1.14	3.390	3.586	99	193	13.1%
Parallel	1.41	3.161	3.351	102	229	13.5%

## Data Availability

The data that support the findings of this study are available from the corresponding authors upon reasonable request.

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
