# Peer review of "Design and Fabrication of 3.5 GHz Band-Pass Film Bulk Acoustic Resonator Filter"

_micromachines, 2024, doi:10.3390/mi15050563_

Round 1

Reviewer 1 Report

Comments and Suggestions for Authors

In this work, BAW devices based on Sc0.2Al0.8N were demonstrated. FBAR showed a large k2 of 13.1%. Then, ladder-type FBAR filters with pass-band of 3.4 to 3.6 GHz were designed and fabricated. The influence of topology was also investigated.

This manuscript is well organized and has sufficient novelty. I will recommend this manuscript for publication in Micromachines if the following points are revised.

1.      In page 1, line 37, authors mentioned that SAW devices is difficult to be used at frequency above 3 GHz for their performance degradation at high frequency. This seems to handled by taking structures like IHP SAW or SUP SAW. If BAW devices have specific advantages over the structures like that? This should be add to the introductions.

2.      Page 5, line 148, there seems to be some minor mistakes in equation 2, “a” is not defined in the article. Is this affect the result of Q, which seems to be too low compared with the common idea?

3.      In Figure 6, authors improved the in-band response by introducing matching elements at ports. The introduction of matching elements may affect the out-of-band rejection at a wider spectrum, and is not discussed in the current version.

4.      With increased orders of ladder-type filter, better out-of-band rejection is achieved with slight influence on IL. The S21 performances change greatly when the order is increased from 3 to 4. This should be explained briefly.

5.      Some minor mistakes in writing like,

“Though the frequency of pass-band is lower than the predicted ones due the fabrication deviation. the pass-band from 3.27 to 3.47 GHz is achieved with a 200 MHz bandwidth and insertion loss lower than 2 dB.” Page 1, line 22.

Most commercially available BAW filters are constructed with film bulk acoustic resonators (FBARs), in which an air cavity is created between the bottom electrode and the carrier wafer.” Page 1, line 40.

Should be modified carefully.

Comments on the Quality of English Language

Some minor mistakes in writing should be modified carefully.

Reviewer 2 Report

Comments and Suggestions for Authors

This paper describes the development and production of ScAlN-based FBAR filters specifically designed for 5G mid-band frequencies. The filters achieve a pass-band range of 3.27 to 3.47 GHz. The paper also includes a thorough analysis of the impact of filter order on out-of-band rejection, insertion loss, and in-band ripple. The findings emphasize the importance of precise thickness control and circuit matching to achieve optimal performance.

Thanks for the well-written manuscript, I have the following questions for this paper:

1.       What is the comparison between the mechanical and thermal stability properties of scandium-doped aluminum nitride and typical materials used in resonators, specifically when exposed to different frequencies and temperatures?

2.       What causes the variations in thickness throughout the manufacturing process, and what specific actions can be taken to manage or reduce these variations in order to ensure that the resonant frequencies roughly match the designed values?

3.       How does the out-of-band rejection improve with increased filter order, yet at the same time the insertion loss and in-band ripple become worse? The author may discuss the underlying physical mechanisms or design limits that cause this to happen.

4.       While higher-order filters manage to keep their out-of-band rejection below -20 dB, the author may discuss particular design elements or material attributes that cause the ripple to become more severe and the high-frequency out-of-band rejection to decrease.

5.       Considering that external circuit elements such as capacitors and inductors can effectively decrease the fluctuation within the desired frequency range but do not affect the reduction in signal strength, the author may discuss alternative circuit design approaches or additional components that can be used to tackle the signal strength issue without compromising the rejection of unwanted frequencies or introducing other undesirable consequences.

Comments on the Quality of English Language

The manuscript is well-written but the paper needs some improvement in the introduction and discussion section to make it more clear and concise.

Reviewer 3 Report

Comments and Suggestions for Authors

Overall, this is a detailed work with a high level of completeness.

However, there are some suggestions for improvement:

Introduction needs to be improved by citing recent conference/journal papers to explain why this work is an improvement over others' work.

e.g. higher keff^2 than Moreira et al.?

e.g. how are your performance specs better than that of Giribaldi et al.?

recommend explaining why the shapes of Fig. 1a , Fig. 5(b-d) were chosen: e.g. due to apodization? what is the apodization strategy?

how were the shape of the FBARs chosen? was it from human design using some theory related to the reflection of acoustic waves from the boundary? Relevant literature could be cited.

please mention in figure label: Fig 1(c) is from circuits model or finite element model or experiment?

suggest to enlarge Fig. 4, especially Fig. 4b. don't squeeze 3 sub-figures in 1 row.

Comments on the Quality of English Language

minor language errors.

"...fabrication deviation. the pass-band from..."

should be comma instead of full-stop.

"especially the fltering action of RF filters"

spelling error.
